# Systems serology detects functionally distinct coronavirus antibody features in children and elderly

Kevin J. Selva[1,24], Carolien E. van de Sandt [1,2,24], Melissa M. Lemke [3,24], Christina Y. Lee [3,24], Suzanne K. Shoffner [3,24], Brendon Y. Chua[1], Samantha K. Davis [1], Thi H. O. Nguyen [1], Louise C. Rowntree[1], Luca Hensen [1], Marios Koutsakos[1], Chinn Yi Wong[1], Francesca Mordant [1], David C. Jackson [1], Katie L. Flanagan [4,5,6,7], Jane Crowe[8], Shidan Tosif [9,10,11], Melanie R. Neeland [9,11], Philip Sutton [9,11], Paul V. Licciardi [9,11], Nigel W. Crawford[9,12], Allen C. Cheng[13,14], Denise L. Doolan [15], Fatima Amanat [16,17], Florian Krammer [16], Keith Chappell[18], Naphak Modhiran [18], Daniel Watterson [18], Paul Young[18], Wen Shi Lee [1], Bruce D. Wines [19,20,21], P. Mark Hogarth[19,20,21], Robyn Esterbauer [1,22], Hannah G. Kelly[1,22], Hyon-Xhi Tan[1,22], Jennifer A. Juno [1], Adam K. Wheatley [1,22], Stephen J. Kent [1,22,23], Kelly B. Arnold [3], Katherine Kedzierska [1,25✉] & Amy W. Chung [1,25✉]

The hallmarks of COVID-19 are higher pathogenicity and mortality in the elderly compared to children. Examining baseline SARS-CoV-2 cross-reactive immunological responses, induced by circulating human coronaviruses (hCoVs), is needed to understand such divergent clinical outcomes. Here we show analysis of coronavirus antibody responses of pre-pandemic healthy children ($n = 89$), adults ($n = 98$), elderly ($n = 57$), and COVID-19 patients ($n = 50$) by systems serology. Moderate levels of cross-reactive, but non-neutralizing, SARS-CoV-2 antibodies are detected in pre-pandemic healthy individuals. SARS-CoV-2 antigen-specific Fcγ receptor binding accurately distinguishes COVID-19 patients from healthy individuals, suggesting that SARS-CoV-2 infection induces qualitative changes to antibody Fc, enhancing Fcγ receptor engagement. Higher cross-reactive SARS-CoV-2 IgA and IgG are observed in healthy elderly, while healthy children display elevated SARS-CoV-2 IgM, suggesting that children have fewer hCoV exposures, resulting in less-experienced but more polyreactive humoral immunity. Age-dependent analysis of COVID-19 patients, confirms elevated class-switched antibodies in elderly, while children have stronger Fc responses which we demonstrate are functionally different. These insights will inform COVID-19 vaccination strategies, improved serological diagnostics and therapeutics.

A full list of author affiliations appears at the end of the paper.

Since the first reported coronavirus disease 2019 (COVID-19) patient in December 2019[1], the severe acute respiratory syndrome coronavirus 2 (SARS-CoV-2) has become a global pandemic, infecting millions of individuals worldwide[2]. Though the majority of COVID-19 patients experience mild symptoms, approximately 20% of cases have more severe disease outcomes involving hospitalization or intensive care treatment, especially in those with underlying co-morbidities such as diabetes and cardiovascular disease[3]. Furthermore, COVID-19-related morbidity and mortality is significantly higher in the elderly population and almost absent in school-aged children[4]. A disproportional outcome in disease severity with increasing age is not unique to the SARS-CoV-2 pandemic and has been observed during previous influenza pandemics[5]. Understanding whether baseline pre-existing immunological responses, induced by previous exposure to seasonal coronaviruses, contribute to such differences may provide important insights into the divergent clinical outcomes between children and elderly.

Antibodies (Abs) are a vital component of the immune response with demonstrated importance in the control of most viral pathogens. However, their ability to respond to new pathogens can be largely affected by age. In influenza studies, elderly donors have increased levels of IgG and IgA antibodies directed to a broad range of historic influenza viral strains, but have decreased ability to generate de novo antibodies towards novel influenza viruses[6]. In comparison, children seem to benefit from more promiscuous antibody responses, better equipped to deal with novel viruses in general[7]. Apart from playing a key role in virus neutralization, Abs also have the capacity to engage Fc Receptors (FcRs) or complement to induce a range of Fc-effector functions, including Ab-dependent cellular cytotoxicity (ADCC), Ab-dependent cellular phagocytosis (ADCP) and Ab-dependent complement activation (ADCA)[8] among others. Abs eliciting Fc-mediated functions are not limited to targeting just neutralizing viral epitopes, such as the SARS-CoV-2 receptor binding domain (RBD), but may instead utilize any available epitope derived from viral proteins[8]. Indeed, non-neutralizing Abs have been shown to be protective against various virus infections by mediating Fc-effector functions[9,10]. A previous SARS-CoV (also called SARS-CoV-1) study associated ADCP with viral clearance[11]; individuals expressing a higher affinity FcγRIIa (CD32a)-H131 polymorphism, associated with enhanced Fc functions, had better disease outcomes[12,13]. A recent study from Schäfer et al. also demonstrated that the loss of Fc-effector function in mice models significantly impaired the potency of several protective anti-SARS-CoV-2 antibodies in vivo. However, Fc functional Abs may also enhance infection or pathology through Ab-dependent enhancement (ADE), previously observed in some SARS-CoV-1 animal vaccine and in vitro studies[14,15]. ADE has the potential to turn mild infections into life-threatening conditions, as exemplified by dengue virus infections, in which non-neutralizing cross-reacting antibodies can exacerbate disease progression[16]. Although cross-reactive antibodies to SARS-CoV-2 antigens, such as the nucleoprotein (NP) and spike 2 (S2), have been detected in uninfected individuals, it is not understood whether ADE could contribute to poorer disease outcome amongst COVID-19 patients, particularly amongst the elderly[17]. Given the alarming rise in COVID-19 deaths, particularly amongst the elderly, it is imperative to profile the impact of age on CoV antibodies. Here, we present evidence of markedly different Ab signatures between pre-pandemic healthy children and elderly samples. We further contrast the pre-pandemic antibody signatures observed in healthy individuals with those of COVID-19-infected patients, children and elderly, to provide a broader context for the implications of these systems serology signatures for disease outcome and future vaccine development strategies.

## Results

**Distinct systems serology signatures in children versus elderly.** In-depth characterization of cross-reactive SARS-CoV-2 Ab responses in healthy children compared to healthy elderly is needed to understand whether pre-existing human coronavirus (hCoV)-mediated Ab immunity potentially contributes to COVID-19 disease outcome. We designed a cross-reactive CoV multiplex array, including SARS-CoV-2, SARS-CoV-1, MERS-CoV and hCoV (229E, HKU1, NL63) spike (S) and NP antigens (Fig. 1a). We assessed CoV-antigen-specific detector levels of isotypes (IgG, IgA, IgM) and subclasses (IgG1, IgG2, IgG3, IgG4, IgA1, IgA2), along with C1q binding (a predictor of ADCA via the classical pathway) and FcγRIIa (CD32a), FcγRIIb (CD32b) and FcγRIIIa (CD16a) soluble dimer engagement (recombinant dimers which mimic FcγR engagement at the immunological synapse and had been previously shown to correlate with a range of Fc-effector functions[18]) (Fig. 1b). A composite dataset of baseline CoV Ab features (14 CoV antigens × 14 detectors, thus 196 Ab features) was generated for the plasma of 89 children, 98 adults and 57 elderly individuals (Fig. 1c and Supplementary Data 1).

To begin, we compared CoV Ab responses between children and elderly. Accounting for multiple comparisons, we identified that 58 of the 196 (29.6%) antibody features were significantly different between the two age groups (all $p < 0.00037$; Fig. 2a and Supplementary Data 2). Children (orange) were characterized by elevated IgM Ab responses targeting a range of CoV antigens, including several SARS-CoV-2 antigens (S, NP and RBD). Furthermore, SARS-CoV-2 spike-specific Abs engaged the high-affinity FcγRIIa-H131 soluble dimers, previously associated with better disease outcome against SARS-CoV-1[12]. FcγRIIa (CD32a) is found on phagocytes such as macrophages, neutrophils and dendritic cells (DCs) and mediates Ab-dependent functions such as ADCP among others[18]. In contrast, the antibody response in healthy elderly individuals (dark blue) was characterized predominantly by IgA and IgG antibodies directed against a range of CoV antigens, including SARS-CoV-2 S and RBD. In particular, hCoV-specific Abs were found to engage soluble FcγRIIIa dimers. FcγRIIIa (CD16a) can be found on NK cells and phagocytes such as monocytes and macrophages, and mediates Ab-dependent functions such as ADCC and ADCP[18,19].

Vastly different CoV serological signatures between children and elderly were also observed when analysing the same data using systems serology[20] (Fig. 2b, c). To identify the minimal signature of Ab features that best distinguished children from elderly, we performed feature selection (Elastic-Net) followed by a supervised multidimensional clustering analysis (partial least squares discriminant analysis, PLSDA; Fig. 2b). Fifteen Ab features selected by Elastic-Net accurately discriminated between children and elderly (99.1% calibration, 98.6% cross-validation accuracy), with significant separation of children and elderly PLSDA scores across the first latent variable (LV1) on the x-axis ($p < 0.0001$, $t = 21.60$) (Fig. 2c). These data reiterated that children have elevated cross-reactive IgM responses to a range of CoV antigens, including SARS-CoV-2 Abs that engaged FcγRIIa-H131 soluble dimers, which were also detected in the multiple comparison analysis (Fig. 2a). Furthermore, this analysis supported that elderly had elevated IgA and IgG to a variety of CoV antigens, including an IgA1 response to SARS-CoV2 RBD and an IgG2 response to SARS-CoV2 NP. Similar trends were also observed when we visualized the data through unsupervised hierarchical clustering (Supplementary Fig. 1a).

**Primed Ab responses to CoVs increase with repeated hCoV exposures.** A recent study by Edridge et al. confirmed that

**a**

| Pathogen | Protein | Acronym | Vol coupled / 12.5 x 10^6 beads | Source | Cat # | Expression | Tag | Accession # | Amino Acid |
|---|---|---|---|---|---|---|---|---|---|
| SARS-CoV-2 | Spike 1 | SARS2 S1 | 100µg | Sino Biological | 40591-V08H | HEK293 | His | YP_009724390.1 | Val16-Arg685 |
| SARS-CoV-2 | Spike 2 | SARS2 S2 | 100µg | Sino Biological | 40590-V08B | Insect | His | YP_009724390.1 | Ser686-Pro1213 |
| SARS-CoV | Spike | SARS1 S | 100µg | Sino Biological | 40150-V08B1 | Insect | His | AAX16192.1 | Met1-Arg667 |
| HCoV-229E | Spike | HCoV 229E S | 100µg | Sino Biological | 40601-V08H | HEK293 | His | APT69883.1 | Cys16-Asn536 |
| HCoV-HKU1 | Spike | HCoV HKU1 S | 100µg | Sino Biological | 40021-V08H | HEK293 | His | YP_173238.1 | Met1-Arg760 |
| SARS-CoV-2 | Receptor Binding Domain | SARS2 RBD | 49.7µg | Florian Krammer | | HEK293 | His | MN908947.3 | Arg319-Phe541 |
| SARS-CoV-2 | Trimeric S | SARS2 Trimer | 100µg | Adam Wheatley | | HEK293 | His | YP_009724389.1 | Met1-Lys1208 |
| SARS-CoV-2 | SClamp | SARS2 SClamp | 100µg | UQ‡ | | | | | |
| MERS-CoV | SClamp | MERS SClamp | 100µg | UQ‡ | | | | | |
| SARS-CoV-2 | Nucleoprotein | SARS2 NP | 100µg | Sino Biological | 40588-V08B | Insect | His | YP_009724397.2 | Met1-Ala419 |
| SARS-CoV | Nucleoprotein | SARS1 NP | 100µg | Sino Biological | 40143-V08B | Insect | His | NP_828858.1 | Met1-Ala422 |
| MERS-CoV | Nucleoprotein | MERS NP | 100µg | Sino Biological | 40068-V08B | Insect | His | AFS88943.1 | Met1-Asp413 |
| HCoV-229E | Nucleoprotein | HCoV 229E NP | 40µg | Prospec-Tany | SARS-001 | E. Coli | His | Q1HVQ0 | Ile39-Asn389 |
| HCoV-NL63 | Nucleoprotein | HCoV NL63 NP | 40µg | Prospec-Tany | SARS-003 | E. Coli | His | ABE73423 | Gln221-Gln340 |
| C. Tetani* | Tetanus Toxin* | Tetanus | 100µg | Sigma | T3194 | | | | |
| Influenza A H1N1* (A/Cali/07/2009) | Hemagglutinin* | H1Cal2009 | 100µg | Sino Biological | 11085-V08H | HEK293 | His | ACP44189.1 | Met 1-Gln 529 |

\* Tetanus toxin and influenza hemagglutinin were used as positive controls
‡ UQ = University of Queensland

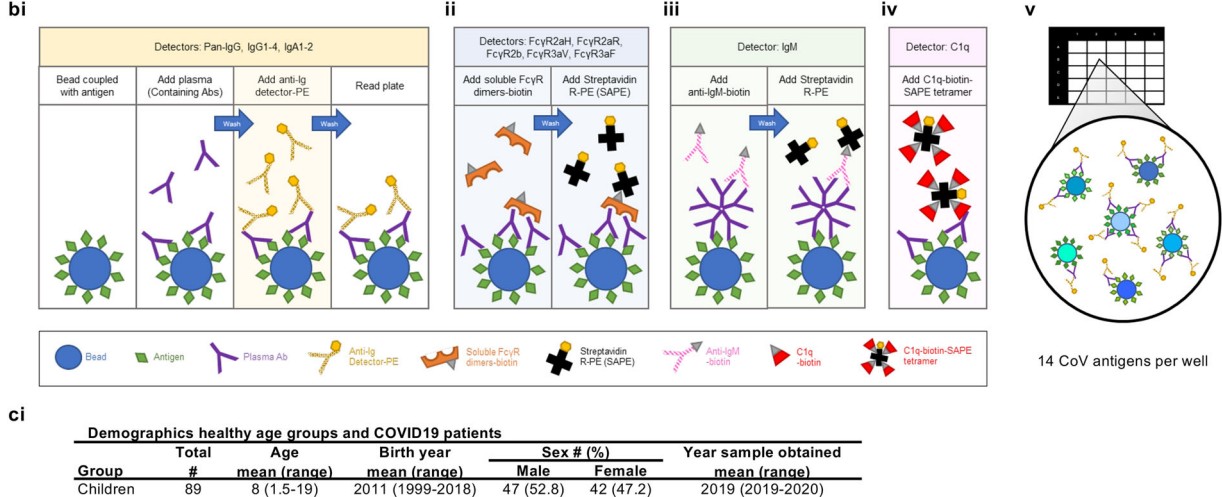

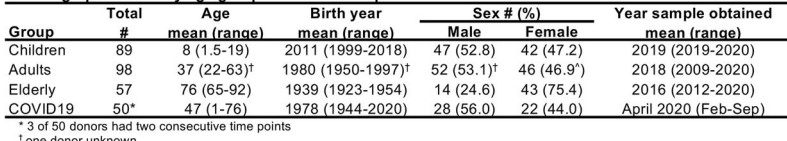

**ci**

**Demographics healthy age groups and COVID19 patients**

| Group | Total # | Age mean (range) | Birth year mean (range) | Sex # (%) Male | Sex # (%) Female | Year sample obtained mean (range) |
|---|---|---|---|---|---|---|
| Children | 89 | 8 (1.5-19) | 2011 (1999-2018) | 47 (52.8) | 42 (47.2) | 2019 (2019-2020) |
| Adults | 98 | 37 (22-63)† | 1980 (1950-1997)† | 52 (53.1)† | 46 (46.9^) | 2018 (2009-2020) |
| Elderly | 57 | 76 (65-92) | 1939 (1923-1954) | 14 (24.6) | 43 (75.4) | 2016 (2012-2020) |
| COVID19 | 50* | 47 (1-76) | 1978 (1944-2020) | 28 (56.0) | 22 (44.0) | April 2020 (Feb-Sep) |

\* 3 of 50 donors had two consecutive time points
† one donor unknown

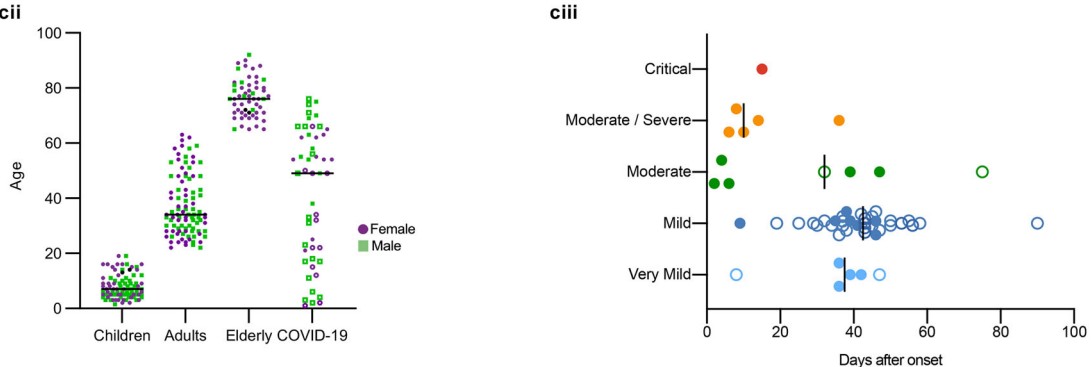

**Fig. 1 Cohort information and setup of the custom CoV multiplex. a** Details of antigens included in the assay. **b** Overview of bead-based multiplex assay. Assay setup for detectors Pan-IgG, IgG1-4, IgA1-2 (**b**-i), FcγR2aH, FcγR2aR, FcγR2b, FcγR3aV, FcγR3aF (**b**-ii), IgM (**b**-iii) and C1q (**b**-iv). FcγR2aH and FcγR3aV are the high-affinity variants of the dimers, while FcγR2aR and FcγR3aF are the respective low-affinity dimer variants. (**b**-v) Beads coupled to respective CoV antigens are added together into wells of a 384-well plate for multiplexing. **c** Overview of the demographics in the healthy donors per age group and COVID-19 patients.

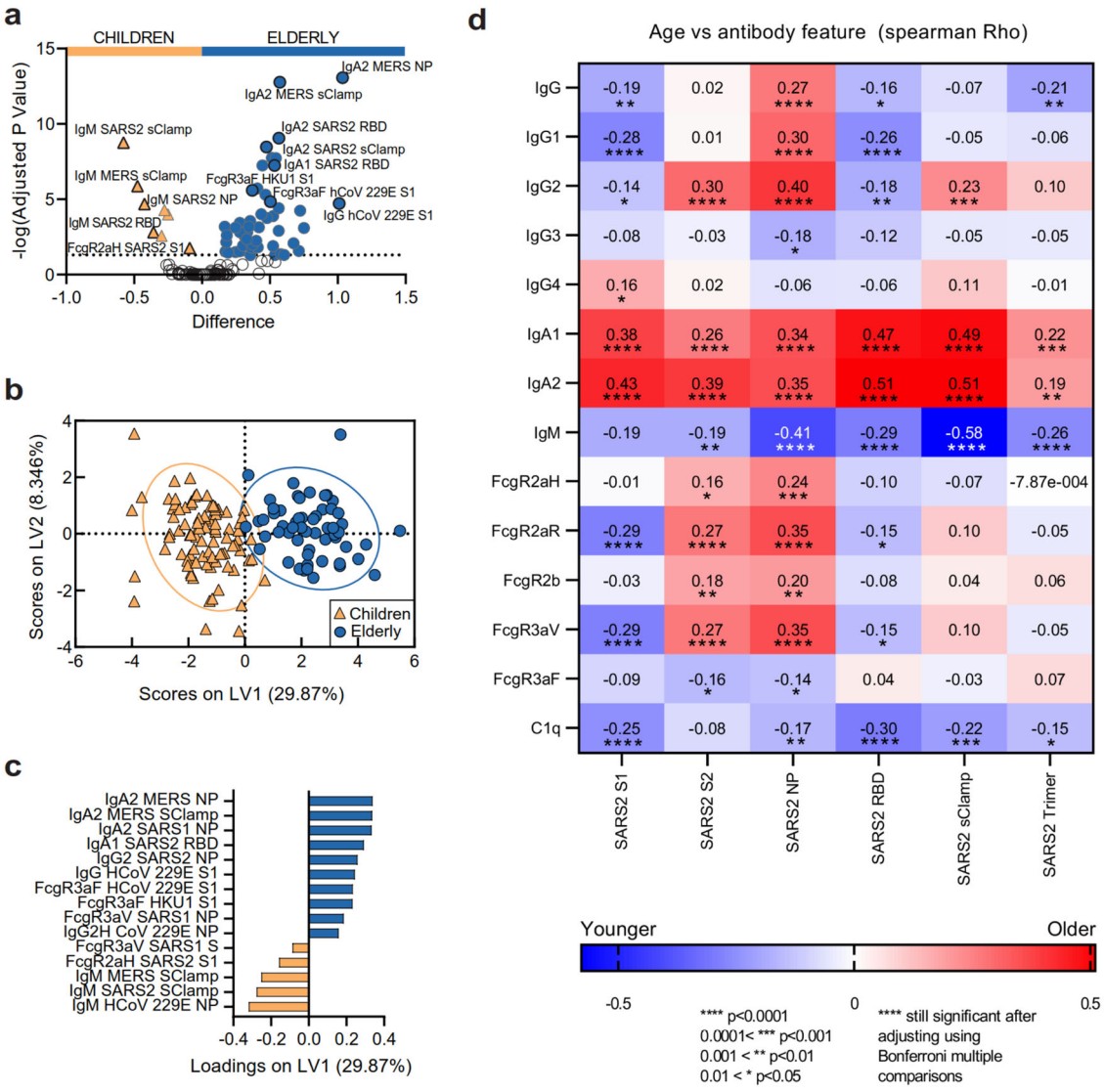

**Fig. 2 Vastly different SARS-CoV-2 serological signatures between healthy children and elderly. a** Volcano plot of healthy children (orange) versus elderly (dark blue), open circles are not significantly different between two groups. Data were z-scored prior to analysis. **b** PCA of all 196 Ab features for healthy children, adults (light-blue square) and elderly. PLSDA scores (**c**) and loadings plots (**d**). Two-tailed Spearman correlation was performed to associate age with the strength of Ab features against the six SARS-CoV-2 antigens. Multiplex assays were repeated in duplicates.

seasonal hCoVs can repeatedly infect individuals of all ages[21], hence the differences between more immature CoV-specific IgM signatures in children and the more mature, class-switched CoV-specific IgA and IgG signatures observed in elderly were likely due to decades of repeated prior exposures to circulating hCoVs in the elderly population. To explore this hypothesis, we investigated whether these differences were gradually introduced in an aging population. A cohort of pre-pandemic healthy adults (ages 22–63) was added to the analysis, together with our children and elderly cohorts. Age was rank-correlated (Spearman's) with the strength of Ab responses picked up by the 14 multiplex detectors against the six SARS-CoV-2 antigens (Fig. 2d). While age once again clearly segregated IgM and IgA responses, we also noticed that Ab responses towards SARS-CoV-2 S2 and NP were largely associated with increasing age. As both S2 and NP[22] are more conserved regions across CoV strains, this observation supports our hypothesis that repeated exposure to circulating hCoV could be driving pre-existing immunity and cross-reactive responses in the elderly. Multivariate regression analysis (partial least square regression, PLSR) and unsupervised hierarchical clustering

performed on the combined cohort, also showed similar associations between CoV Ab responses and age (Supplementary Fig. 1b–d).

**Immune maturation drives mature CoV Ab responses**. To interrogate Ab functionality and cross-reactivity between antigens of selected CoV signatures, we conducted a correlation network analysis, focusing upon significant correlations of the 15 Ab features selected by Elastic-Net. The children's network demonstrates how a range of SARS-CoV-2 S2 Ab features correlate significantly with various features related to SARS-CoV-1 S (Fig. 3a, top left), while features relating to SARS-CoV-2 S1 cluster independently (Fig. 3a, bottom left). This is in line with our previous observations where cross-reactivity of SARS-CoV-2 S1 signatures trended differently from that of S2, possibly due to S2 being more conserved across CoV species[23]. SARS-CoV-2 FcγRIIa was associated with the higher affinity polymorphism-H131 (SARS2 S1 FcgR2aH in Fig. 3a, bottom left) dimer binding against S1, which also correlated strongly with multiple other SARS-CoV-2 Fc responses against S1, including FcγRIIb

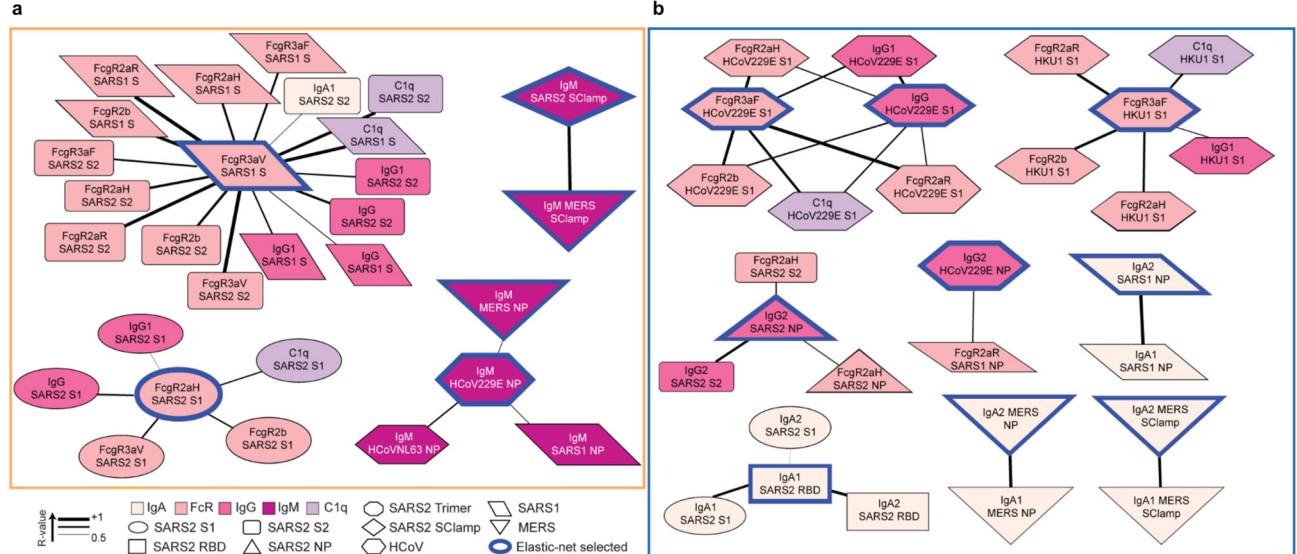

**Fig. 3 Healthy children and elderly have differing correlation networks.** Correlation network analyses for healthy children (**a**) and elderly (**b**) identify features associated with the Elastic-Net-selected features (blue outline). Coded by Ab feature type (colour), antigen (shape) and correlation coefficient (line thickness).

(FcgR2b), C1q and binding to FcγRIIIa-V158 (the higher affinity FcγRIIIa polymorphism, FcgR3aV) dimers. These larger networks suggest that children might have better capacity to engage a range of Fc-effector functions targeting the SARS-CoV-2 S. Separately correlated IgM networks of SARS-CoV-2 antigens together with other CoVs (Fig. 3a, right) suggest that the IgM-dominated immature immune system in children might be more responsive towards SARS-CoV-2 antigens and may be polyreactive in nature, thus might be more rapidly activated upon SARS-CoV-2 infection[7].

The elderly predominantly had hCoV-driven cross-functional Ab signatures to S protein (Fig. 3b, top). Interestingly, while the networks were complex, and driven by both Fc-effector functions and C1q, they were species-specific with a lack in overlap between the networks for either circulating hCoV 229E or hCoV HKU1, unlike the more polyreactive networks observed in the children. These findings further support the notion of pre-conceived immunity in the elderly primed by prior exposures to circulating hCoV drive more focused specificity. Expanding upon our prior observation of matured class-switched cross-reactive antibodies in the elderly, a network of IgA1 and IgA2 responses were formed between SARS2 RBD and S. Similarly, networks involving IgG2 and FcγRIIa-H131 (FcgR2aH) or IgA to CoV antigens were observed (Fig. 3b, bottom). However, notably, these networks were largely species-specific with minimal overlap, suggesting a more rigid immune response present in the elderly, directed towards prior hCoV-exposed antigens. Altogether, the network analyses suggest that children have less exposure to CoV antigens but may have a more adaptable humoral immune responses, both in antigen recognition and Fc responses, targeted towards SARS-CoV-2 compared to elderly. Furthermore, the overall response in children is likely to benefit from the broad polyfunctionality of the SARS-CoV-2 antibody repertoire, which may offer them greater non-neutralizing protection through FcγR engagement than the elderly following initial SARS-CoV-2 exposure, in line with the theory proposed by Carsetti et al.[7].

**HLA class II alleles influence CoV Ab signatures in healthy individuals.** Establishment of an effective humoral immune response after infection and/or vaccination depends on the

generation of affinity-matured long-lived plasma cells and memory B cells, and is correlated with effective activation of T follicular helper ($T_{FH}$) cells[24]. $T_{FH}$ cells are activated by presentation of viral epitopes presented by HLA class II alleles on antigen presenting cells, such as DCs. However, a broad array of HLA class II alleles expressed in humans could affect the activation of $T_{FH}$ cells and thus likely to differentially shape the humoral immune responses. Moreover, several studies demonstrate that variations in HLA class II alleles were associated with susceptibility or resistance to several infectious diseases including MERS-CoV[25,26] and with vaccine-induced Ab responses[27]. Hence, we investigated whether HLA class II alleles could affect CoV Ab signatures observed in healthy individuals, which would improve our understanding on the role of HLA class II alleles in shaping the Ab response upon SARS-CoV-2 infection. We analysed the antibody signatures of healthy individuals for whom HLA class II allele information was available (children $n = 84$, adults $n = 17$ and elderly $n = 10$; Supplementary Fig. 2 and Supplementary Data 1). Data for all healthy donors were pooled since subsets for adult and elderly donors were too small to analyse individually. HLA distributions per age group can be found in Supplementary Fig. 2. To determine whether HLA class II alleles contributed to differences in Ab predisposition, we conducted Elastic-Net and PLSDA to distinguish Ab responses between the two most frequently observed HLA-DQB1, -DRB1, or -DPB1 alleles in our cohort (Supplementary Fig. 2a, d, g). Intriguingly, HLA-DQB1*03:01 and HLA-DQB1*06:02 were associated with distinct Ab features (Supplementary Fig. 2b, c; calibration 86.4%, and 82.6% cross-validation accuracy), and to a minor extent between HLA-DRB1*07:01 and HLA-DRB1*15:01 (Supplementary Fig. 2e, f; calibration 79.1%, and 76.3% cross-validation accuracy), and HLA-DPB1*04:01 and HLA-DPB1*02:01 (82.3% calibration and 72.2% cross-validation accuracy; Supplementary Fig. 2h, i). Discriminating features included both Ab isotypes and FcR engagement for both hCoV, SARS-CoV1 and 2, and MERS antigens. The driving HLA class II allele behind the differences observed between HLA-DQB1*03:01 and HLA-DQB1*06:02, and HLA-DRB1*07:01 and HLA-DRB1*15:01 remains elusive as HLA-DQB1*06:02 and HLA-DRB1*15:01 are strongly co-expressed in our cohort (93.9% of all HLA-DQB1*06:02 donors also expressed HLA-DRB1*15:01;

27.6% of all HLA-typed donors expressed both HLA-DQB1*06:02 and HLA-DRB1*15:01; Supplementary Data 1). Nevertheless, these results suggest that HLA class II alleles, in combination with prior hCoV exposures, can contribute to differences in Ab predispositions in our healthy donor cohort. Hence, HLA class II alleles could contribute to shaping the SARS-CoV-2 Ab response upon infection or vaccination, possibly resulting in differences in SARS-CoV2-specific Ab titres and/or FcR engagements across individuals.

**Distinct Fc Ab signature in COVID-19 patients**. A cohort of 19 SARS-CoV-2 PCR-positive patients (Fig. 1c-iii (closed symbols) and Supplementary Data 3) were screened for SARS-CoV-2 antigen-specific serological profiles (Supplementary Figs. 3 and 4). An individual who was SARS-CoV-2-exposed but remained SARS-CoV-2 PCR-negative was also assessed (Donor D1). Elevated SARS-CoV-2-specific Ab responses in COVID-19 patients relative to healthy or the exposed but PCR-negative individual were observed across multiple titrations (Supplementary Fig. 4). In particular, we found that in the majority of COVID-19 patients, the SARS-CoV-2 antigen-specific Abs bound to FcγRIIIa-V158 and FcγRIIa-H131 soluble dimers at high levels, even at 1:800 plasma titrations, suggesting that besides potential neutralizing activity, alternative antibody-mediated activity such as ADCC and ADCP are likely to contribute to viral clearance[18,19,28].

As the global focus shifts towards serological testing as a strategy for population surveillance to inform government policies, there is an urgent need to distinguish unique Ab profiles in COVID-19 patients to improve the sensitivity and specificity of these tests[29]. Therefore, we next explored if we could detect distinct serological patterns of SARS-CoV-2 antigen-specific Ab features among the various isotypes (IgG, IgA or IgM) that would distinguish our small COVID-19 cohort from healthy individuals (including D1). Through the use of hierarchical clustering, we observed that majority of COVID-19-positive individuals induced high SARS-CoV-2-specific IgM responses, especially to spike antigens, while modest levels of cross-reactive SARS-CoV-2-specific IgM, mainly directed against SARS-CoV-2 NP, were also detected within the healthy individuals, particularly amongst the children (Fig. 4a). Similarly, modest SARS-CoV-2-specific IgA (Fig. 4b) and IgG (Fig. 4c) were observed in healthy, predominantly adult and elderly individuals, though less frequently than IgM, with cross-reactive IgG responses observed at the lowest frequencies of all isotypes (Fig. 4c). Overall, modest levels of SARS-CoV-2 cross-reactive Abs in healthy donors resulted in poor clustering of COVID-19 patients from healthy individuals when a single isotype was assessed, even though multiple SARS-CoV-2 antigens were included. These results suggest that reported low levels of false positives in current serological diagnostics tests could be due to pre-existing levels of cross-reactive Abs that lead to similar serological signatures as observed in SARS-CoV-2-infected individuals when only the quantity of antigen-specific Abs is assessed.

To identify the minimum Ab signature that best distinguishes the COVID-19 patients from healthy individuals, feature selection was used and identified four SARS-CoV-2 Ab variables that differentiated the two groups (Fig. 4d–f), targeting three different SARS-CoV-2 antigens, S trimer, fold on stabilized spike ectodomain, 2P mutation[30], NP, and Sclamp (molecular clamp stabilized spike ectodomain). Intriguingly, antigen-specific engagement of FcγRIIIa-V158 and C1q, but not IgG, was selected. This suggests that SARS-CoV-2 infection potentially induces antigen-specific Ab with distinct Fc qualities, e.g. Fc glycosylation changes, enhancing binding of FcγRIIIa-V158 and C1q[19,20], unlike pre-existing cross-

reactive SARS-CoV-2 Abs observed in our healthy donor cohort. In contrast to previous unsupervised hierarchical clustering for IgA, IgM and IgG (Fig. 4a–c) to multiple SARS-CoV-2 antigens, these four SARS-CoV-2 Ab features had distinct patterns in COVID-19 patients, which led to the clustering of COVID-19 patients together with a single exception, this notably being the healthy, exposed, SARS-CoV-2 PCR-negative individual (Fig. 4d). Strikingly, a supervised PLSDA model of these four features, all associated with COVID-19 patients, could significantly distinguish COVID-19 patients from healthy individuals on LV1 (Fig. 4e, f; *x*-axis, $p < 0.0001$, $t = 34.80$; 98.51% calibration accuracy, 98.51% cross-validation accuracy). Collectively, these results suggest that future COVID-19 serological diagnostic tests could be improved by assessing the Fc quality of antigen-specific Abs in addition to Ab quantity.

To specifically define these four SARS-CoV-2 Ab features, we conducted a correlation network of Ab responses in the COVID-19-positive individuals (Fig. 4g). High levels of correlation were observed between all SARS-CoV-2 spike antigens: S1, S2, RBD, S trimer and Sclamp; while NP antigen-specific Ab features created a separate network. Antigen-specific IgG1 and IgG3, which are the most highly functional IgG subclasses[31] were highly correlated with FcγR and C1q antigen-specific Ab engagement, demonstrating they may be the key mediators of these Fc-effector functions. These results suggest that COVID-19 patients develop a strong SARS-CoV-2-specific Ab response over the course of their infection, eliciting both neutralizing and non-neutralizing functions.

**Qualitative and quantitative differences of IgM in children**. Neutralizing Ab responses targeting the RBD have been observed in the majority of convalescent COVID-19 serum sample, which has been associated with control of SARS-CoV-2[32]. All baseline pre-pandemic samples were tested via a virus microneutralization (VN) assay[33], and not surprisingly, neutralizing antibody titres were solely detected in COVID-19 patients (Supplementary Data 4). However, examining both the quantity and quality of pre-existing RBD may provide insights towards the development of these responses. Baseline RBD isotype-specific levels between healthy children, adults, elderly and COVID-19 patient plasma samples were assessed via multiplex-assay and validated with published ELISA methods[34] (Fig. 5). ELISA data confirmed our earlier observations that children induced elevated IgM (Fig. 5a) while elderly had higher RBD-specific IgA1 responses as measured by multiplex and trended with ELISA IgA results (Fig. 5b), while no differences in IgG were observed among the healthy donors (Fig. 5c). The absence of significant differences between age groups in the ELISA can be accounted for by the fact that we only tested a fraction of the healthy donors and that the multiplex is in general more sensitive.

Since Ab neutralization quality and potency is often correlated with Ab avidity, we conducted a urea disassociation assay on a subset of children, elderly and COVID-19 plasma samples (Fig. 5a-iv, b-iv). Avidity of RBD-specific IgM from children and elderly was significantly weaker than COVID-19 patients ($p = 0.0059$ and $p = 0.0006$, respectively). Interestingly, children's IgM responses span a large and slightly higher range of avidities (median 60.88; IQR 49.37–79.8) as compared to the elderly (median 50.73; IQR 43.83–71.55). No differences in IgA avidity were found between children (median 62.22; IQR 56.37–70.99), elderly (median 63.08; IQR 34.84–76.46) and COVID-19 patients (median 64.77; IQR 32.63–81.78). The avidity assay was not conducted for IgG, due to lack of resolution for healthy individuals in the ELISA. Altogether, our data suggest that compared to the elderly, children may have increased potential to

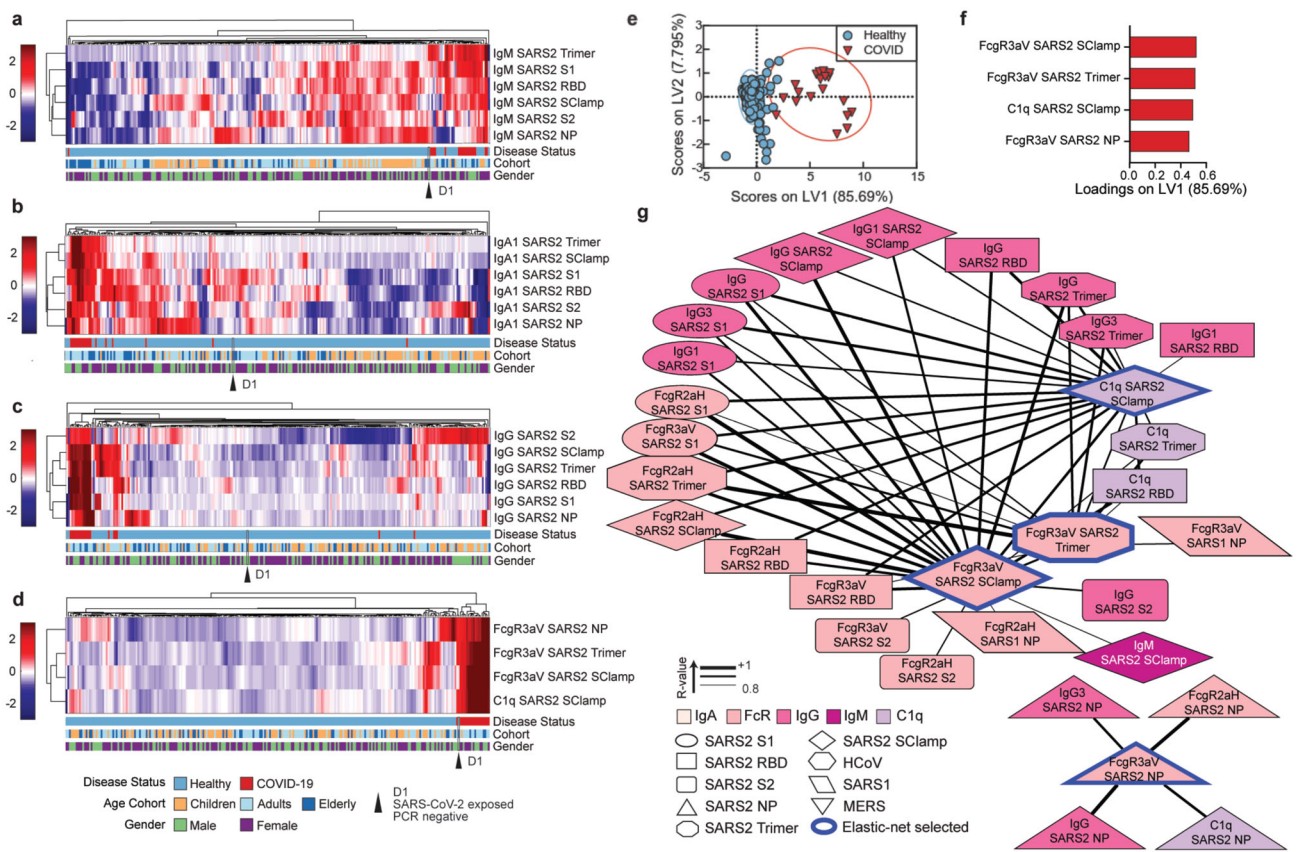

**Fig. 4 Healthy versus COVID-19 serological signatures.** Hierarchical clustering of all SARS-CoV-2 antigens for IgM (**a**), IgA1 (**b**) and IgG (**c**). Levels are coloured from low (dark blue) to high (dark red). Hierarchical clustering (**d**) and PLSDA model scores (**e**) and loadings (**f**) were performed using the four-feature Elastic-Net-selected SARS-CoV-2 antigen signature (98.51% calibration accuracy, 98.51% cross-validation accuracy). Variance explained by each LV is in parentheses. **g** Correlation network analysis for COVID-19 patients was performed to identify features significantly associated with the Elastic-Net-selected features (blue outline). Coded by Ab feature type (colour), antigen (shape) and correlation coefficient (line thickness). Data were z-scored prior to analysis. Multiplex assays were repeated in duplicates.

mount, a higher quality of antibody responses originating from the high binding avidity IgM population which is less commonly observed in the elderly. This may offer them greater development of non-neutralizing protection through FcR engagement than the elderly following initial SARS-CoV-2 exposure, in line with the theory proposed by Carsetti et al.[7].

**Distinct Ab features between children and elderly COVID-19 patients.** In line with a recent study[21], we observed that repetitive infections with seasonal hCoVs have shaped the humoral immune responses in elderly to a more mature and class-switched CoV-specific IgA and IgG response. This is in contrast to the less-experienced antibody signatures observed in children, which potentially could impact the functionality of the humoral immune response following SARS-CoV-2 exposure. Hereto, we investigated differences in age-specific Ab features among COVID-19 patients. To this end, we included additional COVID-19 patients (Fig. 1d-ii-iii; open symbols) to even out distribution across a range of age groups. No significant differences in virus neutralization were observed between the different age groups (Supplementary Data 4, assay 2), indicating that additional immune functions, beyond neutralization may contribute to recovery.

To focus in on the particular serological differences between COVID-19 children and elderly, we performed a COVID-19-specific systems analysis on both cohorts (Supplementary Data 5). Using supervised feature selection followed by multivariate regression analysis, we were able to identify age-dependent Ab signatures in this distinct COVID-19 patient cohort (Fig. 6a, b). We detected that SARS-CoV-2-specific IgA and IgG features were associated with increasing age, especially to both S2[23] and NP[22], which could be a result of cross-reactivity driven by prior exposure to hCoV antigens.

In contrast, SARS-CoV-2-specific functional responses, especially both polymorphisms of FcγRIIa and FcγRIIb to Spike 1 and RBD were associated with COVID-19 cohort children. These data were verified using unsupervised hierarchical clustering in which similar trends were observed (Fig. 6c). To confirm the accuracy of the selected features (Supplementary Fig. 5a), we compared the cross-validation (CV) accuracy of our selected model with randomly selected antibody features, observing significantly higher accuracy ($p = 0.006$) with our model (Supplementary Fig. 5b). Engagement of FcγRIIa is commonly associated with antibody-mediated phagocytosis. Significant univariate differences between Spike 1-specific antibody engagement for both FcγRIIa polymorphisms were observed (FcγRIIaR, $p = 0.0332$; Fig. 6d and FcγRIIaF, $p = 0.0387$; Fig. 6e), despite the small sample size. To further validate that children have enhanced FcγRIIa-mediated antibody functions, we utilized THP-1 monocyte cell line, which express high levels of FcγRIIa and very low levels of FcγRIIIa[35], to assess for antibody-mediated Fc-effector

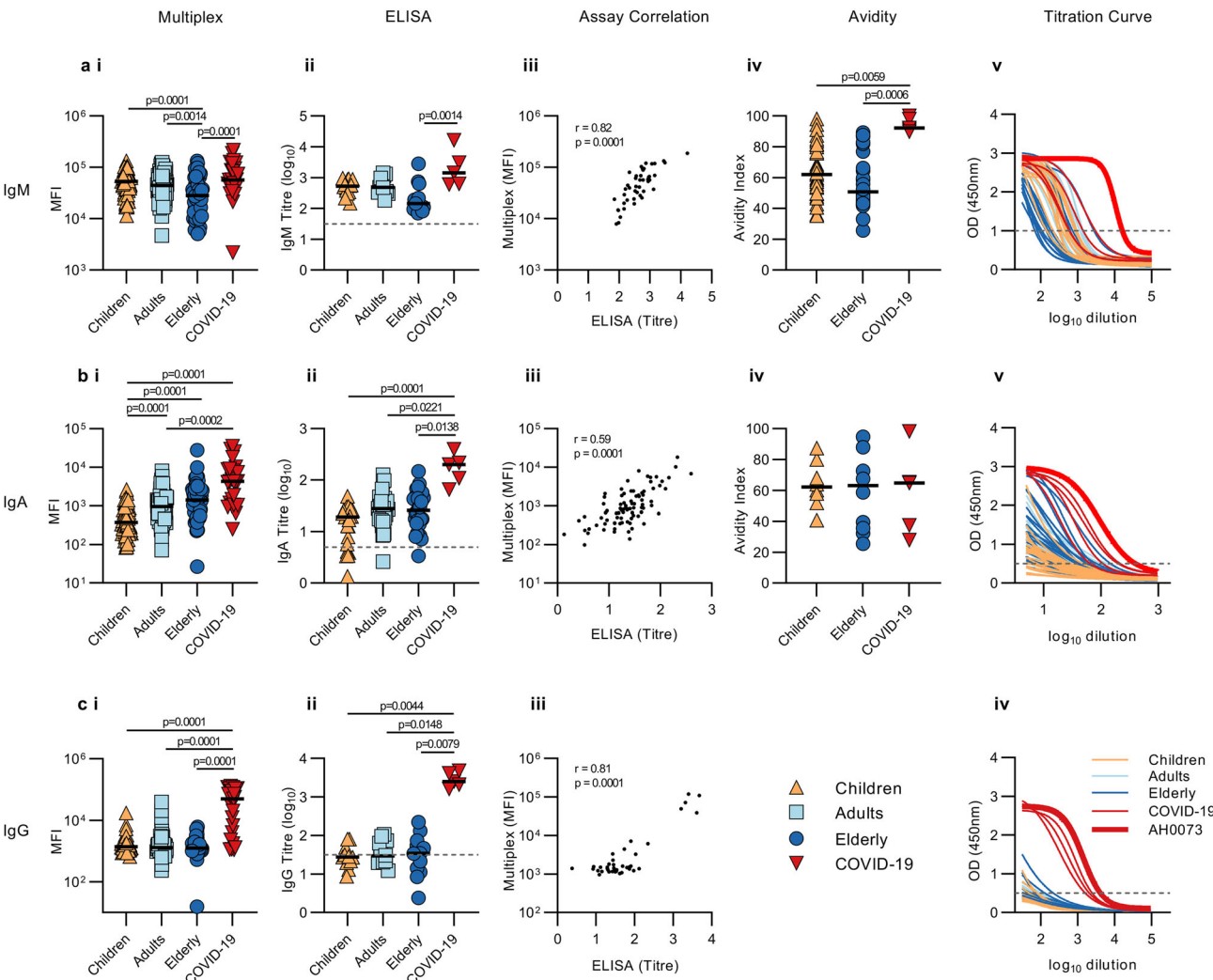

**Fig. 5 Receptor binding domain Abs in healthy versus COVID-19 patients.** Multiplex MFI data for IgM (**a**-i), IgA (**b**-i) and IgG (**c**-i); ELISA endpoint titres for IgM (**a**-ii), IgA (**b**-ii) and IgG (**c**-ii); and their respective correlations (**a–c** iii). Avidity index following urea dissociation for IgM (**a**-iv) and IgA (**b**-iv). Children (orange), adults (light blue), elderly (dark blue) and COVID-19 patients (red). Bar indicates the median response of each group. Statistical significance was determined using one-way ANOVA (Kruskal-Wallis with Dunn's multiple comparisons), exact p-values were provided. Serial dilutions of plasma from healthy children (n = 14) (orange), 12 adults (n = 12) (light blue), 14 elderly (n = 14) (dark blue) and 5 COVID-19 patients (red) tested in IgM (**a**-v), IgA (**b**-v) and IgG (**c**-iv) ELISA. Bold red line represents COVID-19 patient AH0073 who was used as a positive control in all multiplex and ELISA plates. Dashed lines represent cut-offs (15% of positive control for IgA and IgG; 30% for IgM) used to interpolate endpoint titres by non-linear regression analysis. ELISA and multiplex assays were repeated in duplicates.

functions. Children trended to have elevated THP-1 cell-based Fc-mediated uptake of spike-coated beads ($p = 0.1277$, children: median 12.35; IQR 11.09–14.92; elderly: median 10.96; IQR 6.17–13.43; Fig. 6f). Similarly, when using cells transfected with Spike trimer and mOrange as targets, we observed similar trends ($p = 0.0684$, children: median 22.34; IQR 14.55–28.71; elderly: median 14.17, IQR 7.16–21.47; Fig. 6g). Importantly, both cell-based Fc-effector assays highly correlated with each other and to SARS2 S1 FcγRIIa (Fig. 6h). Overall, these findings, albeit done on a small cohort, support our hypothesis that differences in Ab signatures between children and elderly, primed by their prior exposure(s) to circulating hCoV, may contribute to their differential clinical outcomes to COVID-19, where children benefit from their less-experienced immune status prior to SARS-CoV-2 infection, which provides them with the ability to exert a more functional antibody response against SARS-CoV-2.

## Discussion

Using systems serology, we observed distinct coronavirus serological signatures in healthy children compared to elderly. Children had elevated CoV-specific IgM signatures, whereas elderly had more mature, class-switched CoV-specific IgA and IgG, indicating that multiple rounds of infections and/or exposures over several decades might be needed to develop fully experienced CoV humoral immune response. Intriguingly, similarly to the current SARS-CoV-2 pandemic, school-aged children have shown better clinical outcomes during past influenza pandemic outbreaks[5] and can induce more potent, broadly neutralizing Ab responses upon HIV infection[36]. It is plausible that upon infection with SARS-CoV-2, the elderly may preferentially induce skewed Ab responses targeting prior cross-reactive hCoV antigens and as observed in this study, COVID-19-positive elderly induced elevated IgG and IgA antibodies to the more cross-reactive antigens of SARS-CoV2 including S2 and NP, compared to children. Recent Ab repertoire

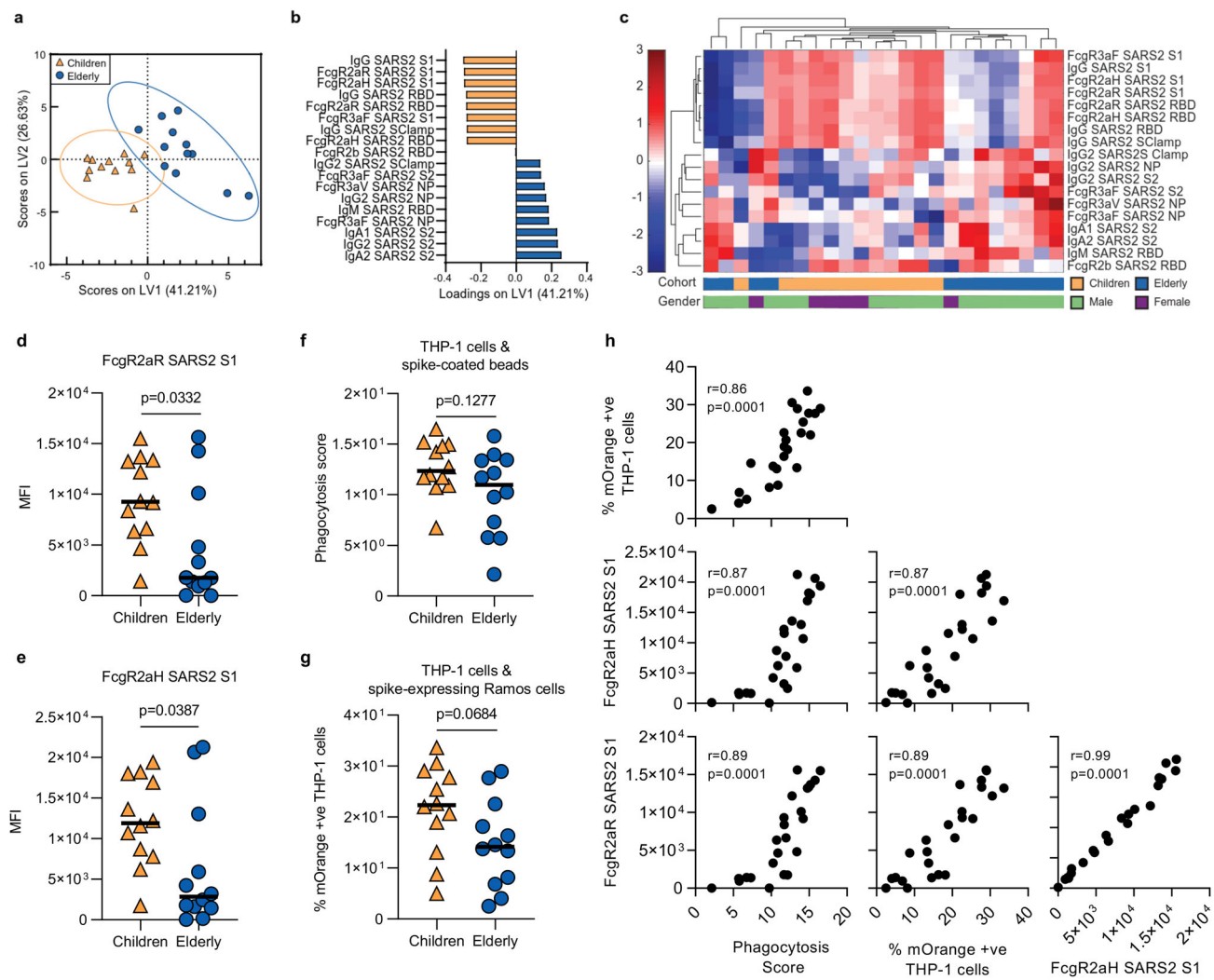

**Fig. 6 COVID-19 Ab responses in COVID-19-positive children and elderly.** PLSDA scores (**a**) and loadings (**b**) plots for the children (*n* = 12) (orange) and elderly (*n* = 12) (dark blue) with an Elastic-Net-selected 18-feature signature (100.00% calibration accuracy, 91.37% cross-validation accuracy). Hierarchical clustering was performed using the 18-feature signature for the children and elderly cohorts (**c**). S1-specific antibody engagement of FcγRIIaR (**d**) and FcγRIIaR (**e**) were amongst the strongest features in the PLSDA loadings plot (**b**) and are significantly elevated in children (Supplementary Data 5 describes all comparisons between children and elderly). THP-1 monocyte cell line antibody-mediated uptake of spike-coated bead assay (**f**) and spike-expressing target cell assay (**g**). Statistical significance was calculated using the two-tailed Mann-Whitney *U* test and exact *p*-values were reported. S1-specific antibody engagement of FcγRIIaR, FcγRIIaR and the two THP-1 SARS-CoV2 spike Fc effector assays highly correlate with each other, as measured by two-tailed Spearman correlation (**h**). Multiplex assays were repeated in duplicates.

study from a single adult COVID-19 subject observed a larger proportion of non-neutralizing Abs that displayed high levels of somatic hypermutation which cross-react with hCoV, which the authors suggest indicate pre-existing memory B cells by prior hCoV[37]. In addition, recent studies have observed boosting of hCoV antibody responses, especially OC43 upon SARS-CoV2 infection[38–40]. As such, upon exposure to SARS-CoV-2 in elderly individuals, the influence of pre-existing memory responses in combination with potentially slower activation of the memory B cell response in general[7], may contribute to them inducing a less effective antibody response.

In contrast, children, who have less-experienced humoral immunity to hCoV, may mount a more targeted immune response towards antigens from SARS-CoV-2, such that FcγRIIa-mediated responses and IgG features targeted against SARS2-CoV-2 S1 and RBD dominated the antibody response in SARS-CoV2-infected children. In line with this observation, we also demonstrate that COVID-19 children responded better than

the elderly using two different antibody-mediated functional assays. These results need to be repeated with in larger cohorts studies; however, these findings give an indication that children may benefit from early Fc-mediated viral clearance and induce more targeted and Fc functional immunity against SARS-CoV-2 antigens in comparison to elderly.

Interestingly, no differences in the in vitro virus neutralization were observed between the different age groups of COVID-19 patients, which indicates that additional immune functions, possibly exerted by the FcR or complement could contribute to the divergent age-related clinical outcomes (Supplementary Data 4). Of note, a recent study comparing neutralizing monoclonal Abs observed that Fc capacity was essential for enhancing protection against SARS-CoV-2 in vivo mouse models, which were not detected via in vitro neutralization assays, as all the currently described neutralization assays lack Fc functions[41]. This highlights the necessity to examine both neutralizing and non-neutralizing responses of Abs in future studies.

Our assay used recombinant FcγR dimers to assess the CoV serological profiles of our healthy donors and COVID-19 patients for additional functions beyond viral neutralization. Recombinant FcγR dimers have previously been demonstrated to be an excellent surrogate system to mimic FcγR engagement at the immunological synapse and was shown to correlate with a range of in vitro cellular Fc effector assays including ADCC and ADCP[19,42]. Similarly, we observed that SARS-CoV2 FcγR dimer engagement strongly correlated with two different cellular Fc effector assays. FcγR dimers have also been used to assess Fc-effector functions against a range of infectious disease pathogens including HIV[43,44], influenza[45,46] and malaria[47]. Similarly, C1q binding has been correlated with in vitro Ab-dependent complement deposition (ADCD) assays[48] and have been used to assess ADCD against HIV[49], Ebola[50] and malaria[51].

We observed distinct SARS-CoV-2 Fc Ab signatures associated with enhanced engagement of the high-affinity FcγRIIIa-V158 dimer and C1q, distinguishing COVID-19 patients from healthy controls. Ab Fc binding to FcγR can be modulated by multiple structural, genetic and post-translational modifications, including Fc glycosylation[52,53]. Fc-effector functions, while beneficial against many pathogens[20,28,54,55], can also enhance infection and pathogenesis in other infectious diseases, including dengue, where disease severity is associated with afucosylated IgG1 that enhances FcγRIIIa affinity[56]. This is also observed with other respiratory diseases including tuberculosis, where greater overall inflammation, including inflammatory Fc glycosylation is associated with poorer disease outcomes[55,57]. IgG with reduced fucosylation has been detected in COVID-19 patients and further implicated with COVID-19 progression, with the critically ill showing aggravated afucosylated-IgG responses against spike, while mild cases displayed higher levels of fucosylation of spike-specific IgG[58,59]. It is important for future larger SARS-CoV-2 serological studies to assess not only quantitative changes in Ab titres, but also qualitative differences in FcR engagements between patients with mild and severe disease, whereas in our study the majority of patients had mild-to-moderate COVID-19.

Pandemic outbreaks provide unique opportunities to study how different aspects of the immune system contribute to the formation of a novel immune response, for example the protective or risk-associated effects of HLA alleles[60]. Due to the low frequency of shared HLA class II alleles in our cohort, this analysis could not be further explored to the impact of the full range of HLA class II alleles, neither to determine the contribution of age nor of sex. However, the distinction in Ab signatures observed in our study emphasizes the need to better understand the contributions of HLA class II alleles to the maturation of humoral immunity and would require a sufficiently large cohort of HLA-typed healthy controls and COVID-19 patients. The current pandemic outbreak provides a unique opportunity to perform such studies, when large patient cohorts which include information on antibody responses, HLA class II phenotypes and disease outcome are combined.

Overall, our in-depth serological profiling of healthy children, elderly and COVID-19 patients brings us closer to understanding why the elderly are more susceptible to COVID-19 and provides insights into Ab Fc signatures associated with convalescence of mild/moderate symptomatic individuals. This knowledge is important for the development of improved serological diagnostics[57], evaluation of convalescent plasma therapeutic trials, and will inform immunogenicity assessment of Ab-based SARS-CoV-2 vaccine strategies, which could potentially extend beyond neutralizing Abs.

## Methods
**Study participants and sample collection.** Our study assessed antibodies to SARS-CoV-2 in a total of 244 healthy individuals and 43 SARS-CoV-2-infected patients (Supplementary Data 1 and 3). Children undergoing elective tonsillectomy (age 1.5–19) were recruited at the Launceston General Hospital (Tasmania) and, apart from fulfilling the criteria for tonsillectomy, they were considered otherwise healthy, showing no signs of immune compromise. Healthy adult donors (age 22–63) were recruited via the University of Melbourne. Healthy elderly donors (age 65–92) were recruited at the Deepdene Medical Clinic (Victoria). All healthy donors were recruited prior to SARS-CoV-2 pandemic. SARS-CoV-2-infected patients (age 1–76) were recruited at the Alfred Hospital (AH), at the Murdoch Children's Research Institute (FFX), by James Cook University (D) and University of Melbourne (CP). Eligibility criteria for COVID-19 acute and convalescent recruitment were having at least one swab PCR-positive for SARS-CoV-2. Each patient was categorized into one of the following 6 severity categories: very mild (stay at home, minimal symptoms), mild (stay at home with symptoms), moderate (hospitalized, not requiring oxygen), severe/moderate (hospitalized with low-flow oxygen), severe (hospitalized with high-flow oxygen) or critical (intensive care unit, ICU). Heparinized blood was centrifuged for 10 min at 300$g$ to collect plasma, which was frozen at −20 °C until required. HLA class I and class II molecular genotyping was performed from genomic DNA by the Australian Red Cross Lifeblood (Melbourne).

Human experimental work was conducted according to the Declaration of Helsinki principles and according to the Australian National Health and Medical Research Council Code of Practice. All donors or their legal guardians provided written informed consent. The study was approved by the Human Research Ethics Committee (HREC) of the University of Melbourne (Ethics ID #1443389.4, #2056761, #1647326, #2056689, #1955465) for healthy adult and elderly donors, Tasmanian Health and Medical HREC (H0017479) for healthy child donors, Alfred Hospital (#280/14) for AH donors, RCH HREC (#63666) for FFX donors, James Cook University (#H7886) for D donors and University of Melbourne (#2056689) for CP donors.

**Deglycosylation of *Escherichia coli*-expressed NP.** To minimize possible bacterial glycosylation background from the *E. coli* expression system, recombinant hCoV 229E and NL63 NP (Prospec-Tany) were first treated with O-glycosidase and PNGase F. Briefly, 40 µg of NP were treated with a cocktail of 8 µl 10X GlycoBuffer 2, 8 µl 10% NP40, 12 µl O-Glycosidase, 12 µl of Remove-iT PNGase F (New England BioLabs) and water for a final volume of 80 µl and incubated at 37 °C for 2 h on a shaker. The respective mixtures were added to Eppendorf tubes containing 100 µl of PSB-washed Chitin magnetic beads (New England BioLabs) to allow the binding and removal of Remove-iT PNGase F. Tubes were agitated for 10 min then placed onto a magnetic separation rack for 5 min. The supernatant was retrieved and passed through a 100 kDa Amicon Ultra centrifugal filter (Merck) to remove remaining O-glycosidase. Finally, NPs were washed with PBS using a 3 kDa Amicon Ultra centrifugal filter (Merck) to prepare them for coupling.

**Coupling of carboxylated beads.** A custom CoV multiplex assay was designed with SARS-CoV-2, SARS-CoV-1, MERS-CoV and hCoV (229E, HKU1, NL63) S and NP antigens, as well as SARS-CoV-2 RBD (gift from Florian Krammer)[34], SARS-CoV-2 Trimeric S (gift from Adam Wheatley) and SClamps of both SARS-CoV-2 and MERS-CoV (gift from University of Queensland) (Fig. 1a). Amino acid sequences for CoV spike and NP are as described in Fig. 1a. Tetanus toxoid (Sigma) and influenza hemagglutinin (H1Cal2009; Sino Biological) were also added to the assay as positive controls, while BSA-blocked beads were included as negative controls. Magnetic carboxylated beads (Bio Rad) were covalently coupled to the antigens using a two-step carbodiimide reaction, in a ratio of 10 million beads-to-100 µg of antigen, with the exception of the deglycosylated NPs mentioned above in which 40 µg were used instead, and SARS-CoV-2 RBD which were used at 49.7 µg instead. Briefly, beads were washed and activated in 100 mM monobasic sodium phosphate, pH 6.2, followed by the addition of Sulfo-*N*-hydroxysulfosuccinimide and 1-Ethyl-3-(3-dimethylaminopropyl) carbodiimide (Thermo Fisher Scientific). After incubation at room temperature (RT) for 30 min, the activated microspheres were washed three times and resuspended in 50 mM MES pH 5.0 (Thermo Fisher Scientific). The respective antigens were added to the activated beads and the mixture was incubated at RT for 3 h on a rotator in the dark. Subsequently, the beads were washed with PBS and blocked with blocking buffer (PBS, 0.1% BSA, 0.02% TWEEN-20, 0.05% Azide, pH 7) for 30 min. Finally, beads were washed in PBS 0.05% Sodium Azide and resuspended as one million beads per 100 µl.

**Luminex bead-based multiplex assay.** The isotypes and subclasses of pathogen-specific antibodies present in the collected plasma were assessed using a multiplex assay as described[61] (Fig. 1b). Using a black, clear-bottom 384-well plate (Greiner Bio-One), 20 µl of working bead mixture containing 1000 beads per bead region and 20 µl of diluted plasma were added per well. From validation experiments in which cross-reactive Abs present in healthy individuals were titrated, an optimal concentration of 1:100 working dilution of plasma was selected for downstream assays (Supplementary Fig. 3a, b). The plate was covered and incubated overnight at 4 °C on a shaker and was then washed with PBS containing 0.05% Tween20 (PBST). Pathogen-specific antibodies were detected using phycoerythrin (PE)-conjugated mouse anti-human pan-IgG, IgG1-4, IgA1-2 (Southern Biotech), at 1.3 µg/ml, 25 µl per well. After incubation at RT for 2 h on a shaker, the plate was washed before the beads were resuspended in 50 µl of sheath fluid. The plate was

then incubated at RT for 10 min on a shaker before being read by the FlexMap 3D. The binding of the PE-detectors was measured to calculate the median fluorescence intensity (MFI). Double background subtraction was conducted, removing first background of blank (buffer only) wells followed by removal of BSA-blocked control bead background signal for each well.

For the detection of IgM, biotinylated mouse anti-human IgM (mAb MT22; MabTech) was added at 1.3 μg/ml, 25 μl per well. After incubation at RT for 2 h on a shaker, the plate was washed, and streptavidin, R-Phycoerythrin conjugate (SAPE, Invitrogen) was added at 1 μg/ml, 25 μl per well. The plate was then incubated at RT for 2 h on a shaker before being washed and read as mentioned above. For the detection of FcγR, soluble recombinant FcγR dimers (higher affinity polymorphisms FcγRIIa-H131, lower affinity polymorphisms FcγRIIa-R131, FcγRIIb, higher affinity polymorphisms FcγRIIIa-V158, lower affinity polymorphisms FcγRIIIa-F158) were provided by Bruce Wines and Mark Hogarth. For the detection of C1q, C1q protein (MP Biomedicals) was first biotinylated (Thermo Fisher Scientific), washed and resuspended in PBS and tertramerized with SAPE. Dimers or tetrameric C1q-PE were added at 1 μg/ml, 25 μl per well, incubated at RT for 2 h on a shaker and then washed. For Dimers, SAPE was added at 1 μg/ml, 25 μl per well, incubated at RT for 2 h on a shaker before being washed and read as mentioned above. Assays were repeated in duplicate. A titration of AH0073 was included in the layout of all multiplex array plates as this patient was known to have IgG, IgM and IgA responses (Supplementary Fig. 4). These titrations were used to normalize replicate multiplex array plates.

**Enzyme-linked immunosorbent assay (ELISA).** Detection of RBD-specific antibodies was performed as described in Stadlbauer et al.[34,62] with the following modifications; Nunc MaxiSorp flat-bottom 96-well plates (Thermo Fisher Scientific) were used for antigen coating, blocking performed with PBS containing 10% BSA and half-logarithmic serial dilutions (beginning at 1:10 for IgA and 1:31.6 for IgG/IgM) performed with PBST containing 5% BSA. For detection of IgG and IgA, peroxidase-conjugated goat anti-human IgG (Fcγ fragment specific; #109-035-098; Jackson ImmunoResearch) or alkaline phosphatase-conjugated rat anti-human IgA (mAb MT20; #3860-9A-1000; MabTech), was used and developed with TMB (Sigma) substrate for IgG or pNPP (Sigma) for IgA. For IgM, biotinylated mAb MT22 (#3880-6-250; MabTech) and peroxidase-conjugated streptavidin (Pierce; Thermo Fisher Scientific) were used. Peroxidase reactions were stopped using 1 M H3PO4 and plates read at 450 or 405 nm on a Multiskan plate reader (Labsystems). All measurements were normalized using a positive control plasma from a COVID-19 patient (AH0073) run on each plate (Fig. 5a–v, b–v, c–v). Endpoint titres were determined by interpolation from a sigmoidal curve fit (all $R^2$ values >0.95; GraphPad Prism 8) as the reciprocal dilution of plasma that produced ≥15% absorbance of the positive control. A total of 28 donors from each cohort was randomly selected for IgA analysis, 14 for IgG and 10–14 for IgM. All assays also included the five same COVID-19 patient samples.

**Antibody avidity assay.** Avidity of antibodies in plasma samples was measured using urea as the chaotropic agent and only performed on samples with detectable RBD-specific antibodies (IgA and IgM). Following incubation of plasma at a 1:10 dilution (IgA) or 1:100 dilution (IgM) on RBD-coated plates, 6 M of urea was added and incubated for 15 min. Bound antibodies were then detected using respective secondary detection reagents described above. The avidity index is expressed as the percentage of remaining antibody bound to antigen following urea treatment compared to the absence of urea.

**Neutralization antibody assay.** SARS-CoV-2 isolate CoV/Australia/VIC01/2020[63] was passaged in Vero cells and stored at −80 °C. Serial two-fold dilutions of heat-inactivated plasma or serum were incubated with 100 TCID50 of SARS-CoV-2 for 1 h and residual virus infectivity was assessed in Vero cells; viral cytopathic effect was read on day 5. The neutralizing antibody titre is calculated using the Reed/Muench method as previously described[64,65].

**Cell-based antibody Fc effector assays.** To examine the COVID-positive plasma of children and elderly for antibody-mediated activation of FcgRIIa expressing THP-1 monocyte cell lines, a previously described bead-based ADCP assay[66,67] was adapted for use in the context of SARS-CoV2[68]. SARS-CoV-2 S trimer was biotinylated using EZ-Link Sulfo-NHS-LC biotinylation kit (Thermo Scientific) with 20 mmol excess according to manufacturer's instructions and buffer exchanged using 30 kDa Amicon centrifugal filters (EMD millipore) to remove free biotin. Biotinylated S was then used to coat the binding sites of 1 μm fluorescent NeutrAvidin Fluospheres beads (Invitrogen) overnight at 4 °C. S-conjugated beads were washed four times with 2% BSA/PBS to remove excess antigen and incubated with plasma (1:100 dilution) for 2 h at 37 °C in a 96-well U-bottom plate. THP-1 monocytes (10,000/well) were then added to opsonized beads and incubated for 16 h under cell culture conditions. Cells were fixed with 2% formaldehyde and acquired by flow cytometry on a BD LSR Fortessa with a high-throughput sampler attachment (HTS). The data were analysed using FlowJo 10.7.1 and a phagocytosis score was calculated as previously described[69] using the formula: (%bead-positive cells × mean fluorescent intensity)/$10^3$. To account for non-specific uptake of S-

conjugated beads, the phagocytosis scores for each plasma sample were subtracted with that of the 'no plasma' control.

The bead-based assay was also adapted to use Ramos cells expressing Spike as target cells[68]. THP-1 monocytes were first stained with CellTrace™ Violet (CTV; #C34557; Life Technologies) as per manufacturer's instructions. In a 96-well V-bottom cell culture plate, Ramos S-orange cells (10,000/well) were incubated with plasma from convalescent or uninfected donors (1:5000 dilution) for 30 min. Opsonized Ramos S-orange cells were then washed prior to co-culture with CTV-stained THP-1 monocytes (10,000/well) for 1 h at 37 °C with 5% CO2. After the incubation, cells were washed with PBS, fixed with 2% formaldehyde and acquired using the BD LSR Fortessa with a HTS. The data were analysed using FlowJo 10.7.1. The percentage of Spike-orange-positive THP-1 monocytes was measured for each plasma sample and background-subtracted with the 'no plasma' control.

**Statistical analysis.** Children versus the elderly Volcano plot was conducted using Prism 8. Statistical significance determined using the Holm-Sidak method, with α = 0.05 adjusted for 196 tests (Ab features). Each feature was analysed individually, without assuming a consistent SD. The overall multiplex dataset was analysed for normal distribution using the Shapiro-Wilk test by Prism 8. The data were further analysed by SPSS statistics 26 (IBM Corp.) using the Kruskal-Wallis one-way analysis with a Bonferroni correction to determine the p-values, differences between groups were considered significant at an adjusted p-value of 0.000035 (Supplementary Data 2). ELISA data were analysed using one-way ANOVA (Kruskal-Wallis one-way analysis with Dunn's multiple comparison) using Prism 8. Comparisons between COVID-positive children and elderly were analysed by two-tailed Mann-Whitney U test (Supplementary Data 5). CV accuracies of randomly selected models were compared to the selected model (Supplementary Fig. 5a, b) based on previously published methods, which use a one-sided Fisher's permutation test to calculate a CV p-value in Matlab[70]. The CV p-value represents the proportion of randomized signatures that outperformed the original feature selected model in cross-validation accuracy.

**Data normalization.** For all multivariate analysis, Tetanus, H1Cal2009 antigens (positive controls) were removed, with the exception of HLA analysis. Any healthy samples with a missing age, or missing Ab features were removed ($n = 9$). When analysing COVID-19 samples with healthy samples, only the features where data were available for all COVID-19 samples were included. COVID-19 samples in the initial cohort lacked entire datasets for IgG4, IgA2, FcγRIIaR131, FcγRIIIaF158 and FcγRIIb, thus these detectors were excluded. The second COVID-19 cohort was normalized separately, contained all detectors and no missing data. When COVID-19 samples were analysed based on the time from disease onset, all visit days were used for each patient. In all other analyses when a patient has two visit days, only the second visit was used. Right shifting was performed on each feature (detector–antigen pair) individually if it contained any negative values, by adding the minimum value for that feature back to all samples within that feature. Following this, all data were log-transformed using the following equation, where $x$ is the right-shifted data and $y$ is the right-shifted log-transformed data: $y = \log 10(x + 1)$. This process transformed the majority of the features to having a normal distribution. In all the subsequent multivariate analyses, the data were furthered normalized by mean centering and variance scaling each feature using the $z$-score function in Matlab. For the HLA analysis, the same data-normalization methods were used, except that positive controls and all samples with any HLA typing were included. Samples with one copy of each most-frequent allele were removed to avoid double classification.

**Feature selection using Elastic Net/PLSR and Elastic Net/PLSDA.** To determine the minimal set of features (signatures) needed to predict numerical outcomes (age, days from symptom onset) and categorical outcomes (age cohort, COVID-19 infection status, HLA allele) a three-step process was developed based on Gunn et al.[71]. First, the data were randomly sampled without replacement to generate 2000 subsets. The resampled subsets spanned 80% of the original sample size, or sampled all classes at the size of the smallest class for categorical outcomes, which corrected for any potential effects of class size imbalances during regularization. Elastic-Net regularization was then applied to each of the 2000 resampled subsets to reduce and select features most associated with the outcome variables. The Elastic-Net hyperparameter, α, was set to have equal weights between the L1 norm and L2 norm associated with the penalty function for least absolute shrinkage and selection (LASSO) and ridge regression, respectively[72]. By using both penalties, Elastic-Net provides sparsity and promotes group selection. The frequency at which each feature was selected across the 2000 iterations was used to determine the signatures by using a sequential step-forward algorithm that iteratively added a single feature into the PLSR (numerical outcome) or PLSDA (categorical outcome) model starting with the feature that had the highest frequency of selection, to the lowest frequency of selection. Model prediction performance was assessed at each step and evaluated by 10-fold cross-validation classification error for categorical outcomes and 10-fold goodness of prediction ($Q^2$) for numerical outcomes. The model with the lowest classification error and highest $Q^2$ within a 0.01 difference between the minimum classification error and the maximum $Q^2$ was selected as the minimum signature. If multiple models fell within this range, the one with the least number of features was selected and if

there was a large disparity between calibration and cross-validation error (over-fitting), the model with the least disparity and best performance was selected.

**PCA**. Principal component analysis (PCA), performed in Eigenvectors PLS toolbox in Matlab, is an unsupervised technique that was used to visualize the variance in the samples based on all of the measured features. Every feature is assigned a loading, the linear combinations of these loadings create a principal component (PC). Loadings and PCs are calculated to describe the maximum amount of variance in the data. Each sample is then scored and plotted using their individual response measurements expressed through the PCs. The percent of variance described by each PC is a measure of the amount of variance in antibody response explained by that respective PC. Separation of groups on the scores plot indicates unsupervised separation of groups based on all features.

**PLSDA**. Partial least squares discriminant analysis (PLSDA), performed in Eigenvectors PLS toolbox in Matlab, was used in conjunction with Elastic-Net, described above, to identify and visualize signatures that distinguish categorical outcomes (age cohort, COVID-19 infection status). This supervised method assigns a loading to each feature within a given signature and identifies the linear combination of loadings (a latent variable, LV) that best separates the categorical groups. A feature with a high loading magnitude indicates greater importance for separating the groups from one another. Each sample is then scored and plotted using their individual response measurements expressed through the LVs. The scores and loadings can then be cross-referenced to determine which features are loaded in association with which categorical groups (positively loaded features are higher in positively scoring groups, etc.). All models go through 10-fold cross-validation, where iteratively 10% of the data is left out as the test set, and the rest is used to train the model. Model performance is measured through calibration error (average error in the training set) as well as cross-validation error (average error in the test set), with values near 0 being best. All models were orthogonalized to enable clear visualization of results. Statistically significant separation of groups on the PLSDA score plots was determined using a two-tailed $t$-test on LV1 scores in Prism 8. Confidence ellipsoids (90% confidence level) were plotted for classification groups by calculating the mean and covariance matrix from the scores data of the first and second LVs (or PCs), which follows a chi-square distribution[73].

**PLSR**. Partial least squares regression (PLSR), performed in Eigenvectors PLS toolbox in Matlab, was used in conjunction with Elastic-Net, described above, to identify and visualize signatures that distinguish numerical outcomes (age, days from symptom onset). This supervised method assigns a loading to each feature within a given signature and identifies the linear combination of loadings (a LV) that best describes the variance in the numerical outcome. As in PLSDA, a feature with a high loading indicates greater importance for describing the variance in outcome. Each sample is then scored and plotted using their individual response measurements expressed through the LVs. The scores and loadings can then be cross-referenced to determine which features are loaded in association with which numerical outcomes (positively loaded features are higher in positively scoring samples, etc.). All models go through 10-fold cross-validation, where iteratively 10% of the data is left out as the test set, and the rest is used to train the model. Model performance is measured through $R^2$ (average goodness of fit in the training set) as well as $Q^2$ (average goodness of prediction in the test set), with values near 1 being best. All models were orthogonalized to enable clear visualization of results.

**Hierarchical clustering**. We visualized separation of numerical (age, days from symptom onset) and categorical (age cohort, COVID-19 infection status) outcomes based on their respective signatures using unsupervised average linkage hierarchical clustering of normalized data. Euclidean distance was used as the distance metric.

**Software**. PCA, PLSDA and PLSR models were completed using the Eigenvector PLS toolbox in Matlab. Hierarchical Clustering and Correlation Networks were completed using MATLAB 2017b (MathWorks, Natick, MA). PCA, PLSDA and PLSR scores and loadings plots were plotted in Prism version. Statistical analysis were performed in SPSS.

**Reporting summary**. Further information on research design is available in the Nature Research Reporting Summary linked to this article.

## Data availability
The coding used for analysis can be found in the Source Coding file. All other data are available from the authors upon request. Source data are provided with this paper.

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

## Acknowledgements

We thank all the participants involved in the study, Daniel Pellicci, Jane Batten and Helen Kent for support with the cohort, and Ebene Haycroft and Brendan Watts for Flexmap3D technical assistance. This work was supported by Jack Ma Foundation to K.K., A.W.C. and A.W., the Clifford Craig Foundation to K.L.F. and K.K., NHMRC Leadership Investigator Grant to K.K. (1173871), NHMRC Program Grant to K.K. (1071916), NHMRC Program Grant to D.L.D. (#1132975), NHMRC Program grant to S.J.K. (#1149990), Research Grants Council of the Hong Kong Special Administrative Region, China (#T11-712/19-N) to K.K., MRFF Award (#2005544) to K.K., S.J.K., A.W.C., J.A., A.K.W. and Emergent Ventures Fast Grant to A.W. A.W.C. is supported by a NHMRC Career Development Fellowship (#1140509), K.K. by NHMRC Senior Research Fellowship (1102792), D.L.D. by a NHMRC Principal Research Fellowship (#1137285). S.J.K. by NHMRC Senior Principal Research Fellowship (#1136322). C.E.S. has received funding from the European Union's Horizon 2020 research and innovation program under the Marie Skłodowska-Curie grant agreement (#792532). L.H. is supported by the Melbourne International Research Scholarship (MIRS) and the Melbourne International Fee Remission Scholarship (MIFRS) from the University of Melbourne. J.A.J. is supported by an NHMRC Early Career Fellowship (ECF) (APP1123673). P.V.L. is supported by a NHMRC CDF2 Fellowship (#1146198). P.S. is supported by DHB Foundation Fellowship. This work is supported by Victorian Government's Medical Research Operational Infrastructure Support Program.

## Author contributions

K.J.S., C.E.S., B.Y.C., T.H.O.N., K.B.A., K.K. and A.W.C. formulated ideas, designed the study and experiments; K.J.S., C.E.S., B.Y.C., S.K.D., T.H.O.N., L.R., L.H., M.K., C.Y.W., F.M., R.E., H.G.K., H.X.T., J.A.J., A.K.W. and A.W.C. performed experiments; F.A., F.K., K.C., N.M., D.W., P.Y., W.S.L., B.W., P.M.H. and A.K.W. contributed unique reagents; J.C., K.L.F., A.C.C., D.L.D., D.C.J., S.J.K., P.V.L., S.T., M.N., P.S., N.C. and K.K. provided unique samples; K.J.S., C.E.S., M.M.L., C.Y.L., S.K.S., B.Y.C. and A.W.C. analysed the experimental data; K.J.S., C.E.S., M.M.L., C.Y.L., S.K.S., B.Y.C., K.K. and A.W.C. wrote the manuscript. All authors reviewed the manuscript.

## Competing interests

The authors declare no competing interests.

## Additional information

[1]Department of Microbiology and Immunology, Peter Doherty Institute for Infection and Immunity, University of Melbourne, Melbourne, VIC, Australia. [2]Department of Hematopoiesis, Sanquin Research and Landsteiner Laboratory, Amsterdam UMC, University of Amsterdam, Amsterdam, Netherlands. [3]Department of Biomedical Engineering, University of Michigan, Ann Arbor, MI, USA. [4]Department of Infectious Diseases and Tasmanian Vaccine Trial Centre, Launceston General Hospital, Launceston, TAS, Australia. [5]School of Health Sciences and School of Medicine, University of Tasmania, Launceston, TAS, Australia. [6]Department of Immunology and Pathology, Monash University, Melbourne, VIC, Australia. [7]School of Health and Biomedical Science, RMIT University, Melbourne, VIC, Australia. [8]Deepdene Surgery, Deepdene, VIC, Australia. [9]Infection and Immunity, Murdoch Children's Research Institute, Melbourne, VIC, Australia. [10]Department of General Medicine, Royal Children's Hospital Melbourne, Melbourne, VIC, Australia. [11]Department of Paediatrics, University of Melbourne, Melbourne, VIC, Australia. [12]Immunisation Service, Royal Children's Hospital Melbourne, Melbourne, VIC, Australia. [13]School of Public Health and Preventive Medicine, Monash University, Melbourne, VIC, Australia. [14]Infection Prevention & Healthcare Epidemiology Unit, Alfred Health, Melbourne, VIC, Australia. [15]Centre for Molecular Therapeutics, Australian Institute of Tropical Health & Medicine, James Cook University, Cairns, QLD, Australia. [16]Department of Microbiology, Icahn School of Medicine at Mount Sinai, New York, NY, USA. [17]Graduate School of Biomedical Sciences, Icahn School of Medicine at Mount Sinai, New York, NY, USA. [18]School of Chemistry and Molecular Bioscience, University of Queensland, Brisbane, QLD, Australia. [19]Immune Therapies Group, Burnet Institute, Melbourne, VIC, Australia. [20]Department of Clinical Pathology, University of Melbourne, Melbourne, VIC, Australia. [21]Department of Immunology and Pathology, Central Clinical School, Monash University, Melbourne, VIC, Australia. [22]ARC Centre of Excellence in Convergent Bio-Nano Science and Technology, University of Melbourne, Melbourne, VIC, Australia. [23]Melbourne Sexual Health Centre, Department of Infectious Diseases, Alfred Health, Central Clinical School, Monash University, Melbourne, VIC, Australia. [24]These authors contributed equally: Kevin J. Selva, Carolien E. van de Sandt, Melissa M. Lemke, Christina Y. Lee, Suzanne K. Shoffner. [25]These authors jointly supervised this work: Katherine Kedzierska, Amy W. Chung. ✉email: kkedz@unimelb.edu.au; awchung@unimelb.edu.au

