## [Peer Review File. · Nature Communications]

REVIEWER COMMENTS

Reviewer #1 (Remarks to the Author):

This manuscript by Selva, van de Sandt, Lemke, Lee, and colleagues uses a multiplexed assay to measure 196 features of the humoral immune response to pandemic and seasonal coronavirus antigens in 244 individuals representing pediatric, adult, and elderly cohorts of uninfected and COVID-19 patients. The authors use a sophisticated systems serology approach and data modeling to find an association and signature that can segregate the different cohorts. One of the main findings centers around the pediatric cohort signature of an IgM, FcγRIIb, and C1q response versus adult/elderly cohorts that display signatures of an IgG, IgA, FcγRIIIaV158, and FcγRIIaH131 response. These findings suggested that pediatric cohorts might have more promiscuous or versatile humoral response that can engage d while adult/elderly cohorts were predisposed to a humoral immune response to conserved coronavirus residues.

While the data is largely suggestive and leads to conjectures it is technically advanced and the authors have strived to address all of the concerns raised by the reviewers and the paper is suitable for publication in *Nature Communications*. This will be a good addition to the current COVID-19 literature base.

Reviewer #2 (Remarks to the Author):

Unfortunately the authors have not addressed concerns from the prior reviews. Two rounds of reviewers have now taken time to provide feedback and yet the original problems remain. The conclusions are speculative and overinterpreted. One of the problematic areas involves the interpretation of Fc receptor binding data which is simplistic to a point that I believe it is detrimental to the field. This basic problem was clearly identified in the first review (Reviewer 3) and addressed in the second review (Reviewers 2 and 3). Yet the problem remains. The speculations around original antigenic sin are also unwarranted, as was pointed out in prior reviews. The same problems with conclusions around the antibody response in children remain. Overall, I cannot recommend this study for publication.

We appreciate the opportunity to re-submit our manuscript to *Nature Communications* and are happy to make any additional modifications suggested by the Editor and Reviewers.

Reviewer #1

(Remarks to the Author)

his manuscript by Selva, van de Sandt, Lemke, Lee, and colleagues uses a multiplexed assay to measure 196 features of the humoral immune response to pandemic and seasonal coronavirus antigens in 244 individuals representing pediatric, adult, and elderly cohorts of uninfected and COVID-19 patients. The authors use a sophisticated systems serology approach and data modeling to find an association and signature that can segregate the different cohorts. One of the main findings centers around the pediatric cohort signature of an IgM, FcγRIIb, and C1q response versus adult/elderly cohorts that display signatures of an IgG, IgA, FcγRIIIaV158, and FcγRIIIaH131 response. These findings suggested that pediatric cohorts might have more promiscuous or versatile humoral response that can engage d while adult/elderly cohorts were predisposed to a humoral immune response to conserved coronavirus residues.

While the data is largely suggestive and leads to conjectures it is technically advanced and the authors have strived to address all of the concerns raised by the reviewers and the paper is suitable for publication in *Nature Communications*. This will be a good addition to the current COVID-19 literature base.

We thank the Reviewer for their constructive comments, appreciation for the importance of the study and the positive endorsement for publication.

As advised by the Reviewer, we have now made significant additional changes to the text, carefully rephrasing many of our statements to remove conjectures such as our statement about original antigenic sin, and only have discussed the enhanced cross-reactive CoV antibody responses in COVID-19+ elderly, as observed in our Results and the recent literature^{1, 2, 3}.

In addition, we have now conducted numerous additional experiments to support our observations, including recruiting additional COVID-19+ children and reassessing them for their CoV antibody responses via multiplex, along with conducting cell-based assays to support our observation that children have more functional Fc effector responses in comparison to elderly (**new Figure 6**).

All these additional modifications have now been highlighted in the resubmitted manuscript (in yellow)

Reviewer #2

Unfortunately the authors have not addressed concerns from the prior reviews. Two rounds of reviewers have now taken time to provide feedback and yet the original problems remain. The conclusions are speculative and overinterpreted.... The speculations around original antigenic sin are also unwarranted, as was pointed out in prior reviews.

We have seriously considered the Reviewer's comments about the original antigenic sin and, as advised, we have now removed all references to original antigenic sin. Instead, we now only discuss in **Lines 391-410** the elevated cross-reactive antibody responses observed in elderly and highlight other recent studies^{1,2,3} that have observed back boosting of hCoV spike antibody responses, especially to OC43, but have carefully refrained from over interpreting the consequences of these cross reactive hCoV antibody responses that are elicited upon SAR-CoV2 infection.

Our Discussion section in **Lines 391-401** now state:

It is plausible that upon infection with SARS-CoV-2, the elderly may preferentially induce skewed Ab responses targeting prior cross-reactive hCoV antigens and as observed in this study, COVID-19 positive elderly induced elevated IgG and IgA antibodies to the more cross-reactive antigens of SARS-CoV2 including S2 and NP, compared to children. Recent Ab repertoire study³⁷ from a single adult COVID-19 subject observed a larger proportion of non-neutralizing Abs, that displayed high levels of somatic hypermutation which cross-react with hCoV, which the authors suggest indicate pre-existing memory B cells by prior hCoV³⁷. In addition, recent studies have observed boosting of hCoV antibody responses, especially OC43 upon SARS-CoV2 infection^{38, 39, 40}. As such, upon exposure to SARS-CoV-2 in elderly individuals, the influence of pre-existing memory responses in combination with potentially slower activation of the memory B cell response in general⁷, may contribute to them inducing a less effective antibody response.

In contrast, children, who have less experienced humoral immunity to hCoV, may mount a more targeted immune response towards novel antigens from SARS-CoV-2, such that FcγRIIIa-mediated responses and IgG features targeted against SARS2-CoV-2 S1 and RBD dominated the antibody response in SARS-CoV2-infected children. In line with this observation, we also demonstrate that COVID-19 children responded better than the elderly using two different antibody-mediated functional assays. These results need to be repeated with in larger cohorts studies, however, these findings give a first indication that children may benefit from early Fc-mediated viral clearance and induce more targeted and fc functional immunity against novel SARS-CoV-2 antigens in comparison to elderly.

One of the problematic areas involves the interpretation of Fc receptor binding data which is simplistic to a point that I believe it is detrimental to the field. This basic problem was clearly identified in the first review (Reviewer 3) and addressed in the second review (Reviewers 2 and 3). Yet the problem remains

We have now conducted two different cell-based Fc functional assays on the COVID-19+ children and elderly (Fig 6f-g), utilizing THP-1 monocyte cell lines, which have previously been described to express high levels of FcγRII (both FcγRIIa and FcγRIIb) but have very low levels of FcγRIIIa (Fleit et al J Leukoc Biol 1991), which make them an ideal monocyte cell line to assess FcγRIIa mediated antibody responses, which we observed are elevated in children compared to elderly (Fig 6d-e). Importantly, we demonstrate that both cell-based assays highly correlate with FcγRII dimer binding. More specifically, we utilised a previously published assay^{6,7} that assesses antibody mediated phagocytosis of Spike Trimer coated beads by THP-1 cells, and demonstrate that this assay highly correlated to spike trimer specific antibody binding to both polymorphisms of FcγRIIa dimers (FcγRIIaH: Spearman $r=0.84$, $p<0.0001$, FcγRIIaR: $r=0.90$ $p<0.0001$). We also observed similar highly significant correlations, utilising Ramos cells transfected with Spike Trimer as targets instead of Spike Trimer coated beads (FcγRIIaH: Spearman $r=0.87$, $p<0.0001$, FcγRIIaR: $r=0.89$ $p<0.0001$). Thus, we clearly illustrate that FcγR dimer binding is highly indicative of FcγR engagement required for cellular function and in concordance with previously published studies in HIV, malaria, influenza and TB, strongly correlates with in vitro cellular Fc effector assays.

These results are detailed in **Fig 6d-h** and the FcγR dimers are discussed in **lines 422-432**

Our assay used recombinant FcγRs dimers to assess the CoV serological profiles of our healthy donors and COVID-19 patients for additional functions beyond viral neutralization. Recombinant FcγRs dimers have previously been demonstrated to be an excellent surrogate system to mimic FcγR engagement at the immunological synapse and was shown to correlate with a range of in vitro cellular Fc effector assays including antibody dependent cellular cytotoxicity (ADCC) and antibody dependent cellular phagocytosis (ADCP)^{19, 41}. Similarly, we observed that SARS-CoV2 FcγRs dimer engagement strongly correlated with two different cellular Fc effector assays. FcγR dimers have also been used to assess Fc effector functions against a range of infectious disease pathogens including HIV^{42, 43}, influenza^{44, 45}, and malaria⁴⁶. Similarly, C1q binding has been correlated with in vitro antibody dependent complement deposition (ADCD) assays⁴⁷ and have been used to assess ADCD against HIV⁴⁸, Ebola⁴⁹ and malaria⁵⁰.

The same problems with conclusions around the antibody response in children remain. Overall, I cannot recommend this study for publication.

We appreciate the Reviewer's concerns, thus we have now recruited additional 7 COVID-19+ children, such that our study now assesses 12 COVID-19+ children, which taking into account the study location, Australia, where many states currently have eliminated COVID-19, this is a significant cohort size. Upon assessment of this extended children cohort, in comparison to elderly, we observed very similar antibody signatures to our preliminary analysis, where elderly had elevated IgA and IgG responses, especially Spike 2, which is one of the most cross-reactive SARS-CoV2 antigens due to higher sequence similarity to hCoV Spike2.

In comparison, children had enhanced FcγR binding to Spike antigens, in particular elevated FcγR2aR, FcγR2aH and FcγR2b engagement to Spike 1 and RBD. FcγR2a is one of the major

FcγRs that mediates antibody-mediated cellular phagocytosis. Thus, to confirm that children have elevated Fc effector functions, we conducted two different cell-based Fc functional assays on the COVID-19+ children and elderly. We observed that children trended to have elevated cell-based Fc effector response utilizing our Spike coated bead-based assay ($p=0.1277$, median 12.35; IQR 11.09-14.92) and Spike transfected target cell assay ($p=0.0684$, median 22.34; IQR 14.55-28.71). These results are clearly restricted by small sample size and need to be confirmed in larger cohort studies (as discussed in **lines 408-410**), however we strongly argue that our study combining both biophysical antibody assessment by multiplex and now two different cell based functional assays, is highly suggestive that children are able to mediate stronger Fc effector responses in comparison to elderly, that predominately induced IgA and IgG responses that targeted more cross reactive antigens of SARS-CoV2 Spike.

These results are presented in **Figure 6**, and described in **lines 353-381**

*To focus in on the particular serological differences between COVID-19 children and elderly, we performed a COVID-19-specific systems analysis on both cohorts (**Extended Data Table 5**). Using supervised feature selection followed by multivariate regression analysis, we were able to identify age-dependent Ab signatures in this distinct COVID-19 patient cohort (**Figure 6a-b**). We detected that SARS-CoV-2-specific IgA and IgG features were associated with increasing age, especially to both S2²³ and NP²², which could be result from cross-reactivity driven by prior exposure to hCoV antigens.*

*In contrast, SARS-CoV-2-specific functional responses, especially both polymorphisms of FcγRIIIa and FcγRIIb to Spike 1 and RBD aware associated with COVID-19 cohort children. These data were verified using unsupervised hierarchical clustering in which similar trends were observed (**Figure 6c**). To confirm the accuracy of the selected features (**Extended Data Figure 5a**), we compared the cross validation (CV) accuracy of our selected model with randomly selected antibody features, observing significantly higher accuracy ($p=0.006$) with our model (**Extended Data Figure 5b**). Engagement of FcγRIIIa is commonly associated with antibody mediated phagocytosis. Significant univariate differences between Spike 1 specific antibody engagement for both FcγRIIIa polymorphisms were observed (FcγRIIIaR, $p=0.0332$; **Figure 6d**; FcγRIIIaF, $p=0.0387$; **Figure 6e**), despite the small sample size. To further validate that children have enhanced FcγRIIIa-mediated antibody functions, we utilised THP-1 monocyte cell line, which express high levels of FcγRIIIa and very low levels of FcγRIIIa³⁵, to assess for antibody mediated Fc effector functions. Children trended to have elevated THP-1 cell based Fc mediated uptake of spike coated beads ($p=0.1277$, children: median 12.35; IQR 11.09-14.92; , elderly: median 10.96; IQR 6.17-13.43; **Figure 6f**). Similarly, when using cells transfected with Spike trimer and mOrange as targets, we observed similar trends ($p=0.0684$, children: median 22.34; IQR 14.55-28.71; elderly: median 14.17, IQR 7.16-21.47; **Figure 6g**). Importantly, both cell-based Fc-effector assays*

highly correlated with each other and to SARS2 S1 FcγRIIIa (**Figure 6h**). Overall these findings, albeit done on a small cohort, support our hypothesis that differences in Ab signatures between children and elderly, primed by their prior exposure(s) to circulating hCoV, may contribute to their differential clinical outcomes to COVID-19, where children benefit from their less experienced immune status prior to SARS-CoV-2 infection which provides them with the ability to exert a more functional antibody response against SARS-CoV-2.

1. Prevost, J. *et al.* Cross-Sectional Evaluation of Humoral Responses against SARS-CoV-2 Spike. *Cell Rep Med* **1**, 100126 (2020).
2. Westerhuis, B.M. *et al.* Severe COVID-19 patients display a back boost of seasonal coronavirus-specific antibodies. *medRxiv*, 2020.2010.2010.20210070 (2020).
3. Aydilto, T. *et al.* Antibody Immunological Imprinting on COVID-19 Patients. *medRxiv*, 2020.2010.2014.20212662 (2020).
4. Wec, A.Z. *et al.* Broad neutralization of SARS-related viruses by human monoclonal antibodies. *Science*, eabc7424 (2020).
5. Carsetti, R. *et al.* The immune system of children: the key to understanding SARS-CoV-2 susceptibility? *Lancet Child Adolesc Health* **4**, 414-416 (2020).
6. Ackerman, M.E. *et al.* A robust, high-throughput assay to determine the phagocytic activity of clinical antibody samples. *J Immunol Methods* **366**, 8-19 (2011).
7. Atyeo, C. *et al.* Distinct Early Serological Signatures Track with SARS-CoV-2 Survival. *Immunity* **53**, 524-532 e524 (2020).

REVIEWERS' COMMENTS

Reviewer #1 (Remarks to the Author):

The authors have made considerable effort to respond to the reviews and critiques and I find the current version a significantly improved, focused, and accurate manuscript after all of the revisions. This version is suitable for publication and, as I stated before, will be a good addition to the COVID-19 literature base.

REVIEWERS' COMMENTS

Reviewer #1 (Remarks to the Author):

The authors have made considerable effort to respond to the reviews and critiques and I find the current version a significantly improved, focused, and accurate manuscript after all of the revisions. This version is suitable for publication and, as I stated before, will be a good addition to the COVID-19 literature base. We thank the Reviewer for the kind recognition of our efforts and endorsement of our manuscript.